# Children’s Health Literacy in Relation to Their BMI z-Score, Food Intake, and Physical Activity: A Cross-Sectional Study among 8–11-Year-Old Children in The Netherlands

**DOI:** 10.3390/children9060925

**Published:** 2022-06-20

**Authors:** Jany Rademakers, Marla T. H. Hahnraths, Onno C. P. van Schayck, Monique Heijmans

**Affiliations:** 1Netherlands Institute for Health Services Research (Nivel), P.O. Box 1568, 3500 BN Utrecht, The Netherlands; m.heijmans@nivel.nl; 2Department of Family Medicine, Care and Public Health Research Institute (CAPHRI), Maastricht University, P.O. Box 616, 6200 MD Maastricht, The Netherlands; mth.hahnrahts@maastrichtuniversity.nl (M.T.H.H.); onno.vanschayck@maastrichtuniversity.nl (O.C.P.v.S.)

**Keywords:** health literacy, child, overweight, obesity, BMI, lifestyle, nutrition, physical activity, prevention, health education

## Abstract

Overweight and obesity in children are an increasing public health problem. Health literacy (HL) is a determinant of obesity and body mass index (BMI) rates in adults, but few studies have addressed the impact of children’s own HL on their weight and lifestyle. In this study, we aim to assess the impact of Dutch children’s HL on (1) their BMI z-score, (2) dietary behaviour, and (3) the amount of physical activity (PA) they engage in. A sample of 139 children (age 8–11 years) filled out a digital questionnaire, including an HL measurement instrument and questions regarding their food intake and PA. Furthermore, the height and weight of the children were measured, and background information was collected using a parental questionnaire. Multiple regression revealed a significant positive relation between children’s HL and their PA. No significant association between children’s HL and their BMI z-score or dietary behaviour was found. HL of children in primary school thus has an impact on some aspects of children’s lifestyle, although more research in a larger, more diverse sample is needed to further investigate this.

## 1. Introduction

Overweight and obesity in children are worldwide increasing health problems. Globally, the prevalence of overweight and obesity among children and adolescents (age 5–19 years) has risen from 4% in 1975 to over 18% in 2016. In 2016, over 340 million children and adolescents (age 5–19 years) were overweight or obese [1]. In 2021, 15.5% of Dutch primary school children (age 4–12 years) were overweight, of which 3.6% were obese [2]. Overweight and obesity in children can have serious health consequences. Furthermore, children who are overweight or obese are more likely to also have weight problems in adulthood and to develop illnesses such as diabetes type II, cardiovascular diseases, and musculoskeletal disorders in later life [3,4,5]. Preventing overweight and obesity in childhood is, therefore, an important public health priority. The main causes of overweight and obesity in childhood are an unhealthy diet (intake of food high in fat and sugars and low in vitamins, minerals, and other healthy nutrients) and little physical activity (PA) [3]. In the Netherlands, the majority of children do not lead a healthy lifestyle. Over the period of 2014–2016, only 20% of 9–12-year-old Dutch children consumed at least 200 g of fruit per day, and only 25% consumed the recommended daily 150–200 g of vegetables [6]. In 2020, almost four out of every ten 4–11-year-old Dutch children (39.3%) did not engage in sufficient PA according to the relevant age-appropriate norms [7].

There is a clear association between children’s weight and their socio-economic background; having a low socio-economic status (SES) is one of the most prominent risk factors for developing obesity in Western countries [8,9,10]. This is partly caused by poverty and lack of financial resources, which make a healthy lifestyle less accessible since fresh and healthy food is generally more expensive [11]. Psychosocial aspects such as childhood adversity, family dysfunction, insecurity, chronic stress, and emotional factors also play a role in the unequal prevalence of childhood overweight and obesity across populations [12]. Health literacy (HL) is another important determinant of health inequity. HL can be defined as “people’s motivation, knowledge, and competences to access, understand, appraise, and apply health information to make judgements and take decisions in everyday life concerning healthcare, disease prevention, and health promotion” [13]. There is strong evidence of a social gradient in HL: people with a lower SES more often have limited HL skills [14]. A recent systematic literature review demonstrated that limited HL is a determinant of obesity and higher body mass index (BMI) rates in both children and adult populations [15].

With respect to children’s health outcomes, studies often look at the influence of the HL level of parents and caregivers [16,17]. This is understandable, as especially for young children, parents and caregivers are the persons who make most decisions impacting the life and health of their children. For example, they make decisions regarding grocery shopping, dinner preparation, and enrolment in sports clubs. Evidence suggests that children of parents with high HL, in general, have better health outcomes (e.g., regarding their weight status) than children of parents with low HL [15,16]. However, as children grow older, they become more autonomous in their preferences and behaviours regarding food intake and PA. They become more susceptible to the influence of their peers and to marketing efforts aimed at promoting certain foods and beverages [18,19]. Peers’ influence on children’s eating behaviour is often found to be negative, leading to an increase in consumption of foods that are energy-dense and have little nutritional value [18]. Furthermore, unhealthy food and beverage marketing through media popular among children can negatively impact diet-related outcomes, such as food choice and intake [19].

Only very few studies have addressed children’s own HL in relation to their weight, food intake, and/or PA [20,21,22]. The scarcity of studies in this domain has been partly due to the lack of appropriate instruments to measure children’s HL [23,24,25]. However, in 2018–2019 a children’s version of the European Health Literacy Survey Questionnaire was developed and tested in Germany [26,27,28]. The instrument, the HLS-Child-Q15, is tailored to fourth graders (9–10 years old) and was subsequently translated and validated in other countries, among which was the Netherlands [29]. In a Dutch sample of 8–11-year-old children (*n* = 209), 17.2% had a low HL score (lowest quintile), 61.1% had a medium score (second to fourth quintile), and 21.7% had a high HL score (fifth quintile). Older children (10–11 years) had significantly higher HL scores compared to younger children (8–9 years). No significant differences in overall mean HL scores were found across sex, ethnicity, and SES [29]. The availability of the HLS-Child-Q15 generates new opportunities to gain insight into the association between children’s own HL and their health status and lifestyle. In the study presented in this article, we aimed to assess the impact of Dutch children’s HL on (1) their BMI z-score, (2) their dietary behaviour (fruit consumption, vegetable consumption, water consumption, and soft drink consumption), and (3) the amount of PA they engage in. It was hypothesised that children’s HL would be negatively correlated with their weight and positively with their health-related behaviours.

## 2. Materials and Methods

### 2.1. Participants

This cross-sectional study is part of a larger research project. This project investigates the effects of school-based health-promoting initiatives on children’s health and well-being. Outcome measures include BMI, dietary, and PA behaviours. The study was conducted in 12 primary schools in Limburg, a province in the south of the Netherlands. Data collection for the present study was part of the overall project’s baseline measurements. All pupils of grades three and four (aged 8–11 years; corresponding to study years five and six in the Netherlands) of the twelve schools (*n* = 436) were eligible to participate in the present study; there were no further inclusion or exclusion criteria. Recruitment for the study was done via brochures for parents, with information on the study aims, procedures, and handling of the data. In addition, researchers went to the classrooms to give information to the children about the project and to encourage them to take part. A special session was organised in which parents could ask questions to the researchers about the project. All children who participated were required to hand in a completed informed consent form, signed by both parents/guardians. The Medical Ethics Committee Zuyderland in Heerlen waived the need for ethical approval of the research project (METC-Z no. METCZ20190144). The project was registered in the ClinicalTrials.gov database on 9 December 2019 (NCT04193410).

### 2.2. Measures

#### 2.2.1. Data Collection Procedures

BMI: Height and weight of all participating children were measured to establish BMI. Children were measured outside at the playground, wearing light clothes and no shoes. All measurements were conducted two times, with a third measurement if the difference between the first two was too large (pre-set limit; weight ≥ 0.2 kg, height ≥ 0.5 cm). Weight was measured to the nearest 0.1 kg (Weighing Scale 803, Seca, Hamburg, Germany), and height was measured to the nearest 0.1 cm (Stadiometer 213, Seca, Birmingham, UK).

Questionnaire: All participating children filled out a digital questionnaire, including questions regarding their diet and PA. At the end of the questionnaire, the HLS-Child-Q15 was included [29]. The questionnaire was filled out in class during class hours and took about 30 min to complete. During questionnaire administration, a minimum of one member of the research team was present in the classroom. Due to the digital questionnaire’s nature, participants could not skip questions, resulting in no true missings in the collected data. However, participants could select the “do not know” option when answering the HLS-Child-Q15, and no mean HL scores were calculated for respondents with >3 “do not know” responses (maximum missing rate of 20%).

#### 2.2.2. Covariates

Data on children’s age and gender were collected via the school’s administrative database. Through a (digital) parental questionnaire, data on children’s SES and ethnicity were collected. SES was calculated as the mean of standardised scores on the educational level of both parents. The mean scores were categorised into low, middle, and high SES (based on tertiles). Children’s ethnicity was determined by the parental country of birth and divided into (1) Dutch, (2) Western (i.e., all other European countries (excluding Turkey), North America, Japan, Indonesia, Oceania), and (3) non-Western. If at least one parent was born in a Western (other than the Netherlands) or non-Western country, the child’s ethnicity was labelled Western or non-Western, respectively.

#### 2.2.3. Outcomes

BMI z-score: BMI was assessed by combining weight and height. BMI z-scores (BMI scores adjusted for child age and sex) were calculated by using Dutch reference values [30,31].

Dietary behaviour: To assess children’s dietary behaviour, four questions on the frequency of fruit, vegetable, water, and soft drink consumption were asked. Questions were formulated in the following way: “How often do you normally consume fruit/vegetables/water/soft drinks?” with response options (1 = never, 2 = almost never, 3 = sometimes (1–3 times per week), 4 = often (4–6 times per week), 5 = every day).

PA behaviour: The digital questionnaire contained ten items from the International Physical Activity Questionnaire for Children (IPAQ-C). Each item resulted in an activity score ranging from 1 to 5, and the mean of these scores was calculated to obtain the total activity summary score (ranging from 1 (lowest PA) to 5 (highest PA)) (4).

HL: HL was measured with the Dutch version of the HLS-Child-Q15, which contains 15 items that assess the child’s perceived ability to find, understand, appraise, and apply health information. All items were phrased as “How easy or difficult is it for you to…”. Responses were given on a four-point Likert scale (response options “very difficult”, “difficult”, “easy”, and “very easy”). Furthermore, a “do not know” response category was incorporated for each item. Higher scores correspond with higher HL (higher perceived ease in dealing with health information).

### 2.3. Statistical Analyses

Analyses were performed using IBM SPSS Statistics for Windows (version 25.0) and Stata 16.1. Participants who selected the “do not know” answer option in the HLS-Child-Q15 > 3 times and/or had missing values on any of the other outcomes (gender, age, grade, SES, BMI z-score, PA score, fruit, vegetable, water, and/or soft drink consumption) were excluded from the analyses. Next to the overall mean HL scores (calculated by dividing the sum score of valid responses by the total number of valid responses), HL was examined using quintiles (first quintile = ‘low HL’, second to fourth quintile = ‘medium HL’, fifth quintile = ‘high HL’).

The variables for children’s mean HL scores and age were centred by subtracting the old variables’ mean from the old variable.

After checking if the data met the necessary assumptions, regression analyses were performed to examine the relationship between mean HL scores/HL quintiles and BMI z-scores, PA summary scores, and fruit, vegetable, water, and soft drink consumption. As participants were nested in classes, which were nested in schools, it was checked whether there was a significant variance in class and/or school level for the various outcome variables. If this was the case, multi-level regression analyses were used. In all other cases, multiple linear regression analyses were used. Children’s age, SES, and gender were included as covariates in all analyses. For all analyses, a two-sided *p*-value ≤ 0.05 was considered statistically significant.

## 3. Results

### 3.1. Sample

Parental consent was obtained for 215 of the 436 students eligible for study participation (49.3%). Thirty-five participants were excluded from analyses due to having selected “do not know” > 3 times, and 41 were excluded due to missing values on SES, resulting in 139 children being included in the present study. Almost half of the sample was male (49.6%), and the mean age of the participating children was 9.7 years (standard deviation (SD): 0.656). The majority had a Dutch background (94.9%) and an SES in the highest tertile (51.1%). Table 1 reports the sample characteristics.

### 3.2. Children’s HL and Their BMI z-Score

No significant variance in class and/or school level was found for children’s BMI z-scores, which is why a multiple regression was run to predict BMI z-score from mean HL, gender, age, and SES. These variables did not significantly predict BMI z-score, F(4, 134) = 0.354, *p* = 0.841, adjusted R^2^ = −0.019. None of the four variables added statistically significantly to the prediction, *p* > 0.05. A second multiple regression was run to predict BMI z-score from HL quintiles, gender, age, and SES. These variables did not significantly predict BMI z-score, F(4,134) = 0.479, *p* = 0.751, adjusted R^2^ = −0.015. None of the four variables added statistically significantly to the prediction, *p* > 0.05 (Table 2).

### 3.3. Children’s HL and Their PA Score

A significant variance on class level (not on school level) was found for children’s PA summary scores (intraclass correlation coefficient = 0.279; *p* = 0.000), which is why multi-level regression analyses were performed to examine the relationship between children’s HL and their PA scores.

These analyses revealed a significant relationship between children’s mean HL score and their PA summary score (B = 0.356; *p* = 0.002) (Table 3). A significant relationship between HL quintiles and PA summary scores was also found, revealing that participants with the lowest HL had significantly lower PA summary scores than children with medium HL (B = −0.499; *p* = 0.000).

### 3.4. Children’s HL and Their Dietary Behaviour

No significant variance on class and/or school level was found for any of the children’s dietary outcomes, which is why multiple regression analyses were performed to predict children’s dietary outcomes from HL (mean HL or HL quintiles), gender, age, and SES. Before conducting the analyses for fruit and vegetable consumption, three outliers were excluded from the dataset. In the regression analyses, including mean HL scores, these variables did not significantly predict fruit consumption, F(4, 131) = 0.612, *p* = 0.654, adjusted R^2^ = −0.012; vegetable consumption, F(4, 131) = 0.911, *p* = 0.459, adjusted R^2^ = −0.003; water consumption, F(4, 134) = 1.18, *p* = 0.322, adjusted R^2^ = −0.005; or soft drink consumption, F(4, 134) = 1.69, *p* = 0.157, adjusted R^2^ = −0.019. None of the four variables added statistically significantly to the predictions, *p* > 0.05. In the regression analyses including HL quintiles, these variables did not significantly predict fruit consumption, F(4, 131) = 0.323, *p* = 0.862, adjusted R^2^ = −0.020; vegetable consumption, F(4, 131) = 0.997, *p* = 0.412, adjusted R^2^ = 0.000; water consumption, F(4, 134) = 1.10, *p* = 0.361, adjusted R^2^ = 0.003; or soft drink consumption, F(4, 134) = 1.46, *p* = 0.218, adjusted R^2^ = −0.013. None of the four variables added statistically significantly to the prediction, *p* > 0.05 (Table 4).

## 4. Discussion

In the present study, we aimed to assess the association of Dutch children’s own HL with (1) their BMI z-score, (2) dietary behaviour, and (3) amount of PA they engage in. The children in our study were 8–11 years old. Our hypothesis was that children’s HL is negatively correlated with their weight and positively with their health-related behaviours.

The first part of our hypothesis was not confirmed. This finding is contrary to two of the three other studies that studied the association between HL and their BMI scores. In total, we identified three studies that focused on primary school children [20,21,22]. Two studies, one in a New York sample of overweight children and adolescents (age 6–19 years) and one in a population-based study of Taiwanese children (age 11–12 years), demonstrated an inverse relation between children’s HL and their BMI—the higher their HL, the less likely children were to be overweight or obese [20,21]. A study in Turkey, however, did not find a significant correlation between children’s HL and their BMI [22]. Unlike the studies in New York and Taiwan [20,21], no significant association between children’s HL and their BMI z-score was found in the present study. This might be due to the fact that in our sample, the variation in BMI z-score was limited: values ranged between 14 and 19 (mean 16.8). This means that the BMI of most participants in our sample was within the normal range for children in this age group [30,31]. Since 15.5% of the Dutch children in primary school are overweight, our sample is not representative in this respect [2].

Concerning the second part of our hypothesis, the results were different for dietary intake and PA. With regard to dietary behaviour, there was no significant association between children’s HL and their vegetable intake, fruit intake, water consumption, or soft drink consumption. With respect to fruit intake, there was little variation in the sample, since almost six out of ten participants reportedly consume fruit every day. Given the fact that in the overall population of 9–12-year-old children, only one in five eats a daily portion of 200 g fruit, our sample seems to be eating healthier in this respect [6].

Children’s HL, however, did show an evident association with the amount of PA they engage in. The present study showed a significant positive association between children’s HL score and their PA score—the higher the HL, the higher the participants’ PA score. Participants with HL scores in the medium or highest quintile had significantly higher PA scores than participants with HL scores in the lowest quintile.

While the association between Dutch children’s HL and their BMI z-score and dietary behaviour in this study was non-significant, possibly due to the homogeneity of the sample, the relationship between their HL and amount of PA was evident. Supposedly, children in this age range (8–11 years) have more autonomy over their PA behaviour than they have over their diet. While their parents or caretakers are most often responsible for shopping, filling their lunch boxes, and diner preparation, children themselves are able to choose whether they want to take part in active play and physical games, both during and after school time. A systematic review by Buja et al. also consistently found a positive association between HL and PA in different age groups [32]. They conclude that ‘individuals with a better-developed HL have skills and capabilities that enable them to engage in various forms of personal health-enhancing behaviour, such as regular PA’ [32]. We already know from other research that there is also a relationship between low parental HL, SES, and some child health behaviours likely to negatively impact their health and well-being, including unhealthy dietary intake [33]. If children in this age group are still heavily dependent on their parents regarding their food and beverage intake, a relevant topic for further research would be to study older children’s lifestyle behaviours in relation to HL level since the autonomy of children and adolescents is likely to increase with age. Furthermore, as HL is hypothesised to develop already at an early age and parents are important role models for young children, it would be interesting to investigate a potential association between parents’ and children’s level of HL. This has not yet been a subject of investigation.

PA is an important aspect of a healthy lifestyle. However, PA alone is not enough to counter overweight and obesity. Most intervention programmes aimed at reducing childhood overweight and obesity use combined strategies to improve both PA levels and food intake [34]. Considering the significant association between children’s HL and their PA level that was demonstrated in the present study, efforts aiming to improve children’s HL might be a valuable addition to interventions aiming to reduce childhood overweight and obesity.

The present study has several strengths and limitations. An important strength of this study is that it is one of the few studies worldwide and the first study in the Netherlands to assess the relation between children’s HL with aspects of their health and lifestyle. The study was conducted using an HL measurement instrument specifically designed for and tested in this age group. It is the first (and only) Dutch-language instrument to measure HL in children.

The most important limitation of the study is the homogeneity of the included sample. All children came from one area in the Netherlands (the province of Limburg), and more than half of the children came from the highest SES tertile. Information about the non-response group is generally lacking, which limits the results’ generalisability. It is likely that the relatively large group of parents who did not consent to their children participating in this research project (50.7%) had a lower SES, thus leading to a sample bias. Further ways to increase the participation of both adults with a lower SES and their children in research should be explored, leading to more inclusive studies and more generalisable study outcomes. The lack of variation in some outcome measures (e.g., BMI z-scores, fruit and vegetable consumption) and the relatively low number of participating children (*n* = 139) might have limited the ability to detect significant differences. Furthermore, the outcome measures with respect to dietary behaviour and PA were subjectively measured (with a questionnaire), which may have led to social desirability bias [35]. However, children were encouraged to give honest answers and confidentiality was stressed during questionnaire administration to minimise this risk of bias. Furthermore, the questions with respect to dietary behaviour were relatively global (e.g., questions regarding consumption frequency instead of consumed amount) and did not cover all possible unhealthy behaviours (e.g., intake of candy and sweets, consumption of crisps, fast food, and sugar). More objective and sensitive ways of determining the intake of food and beverages could generate more precise insights into children’s dietary behaviours. Furthermore, the HLS-Child Q-15 questionnaire has only been used once (in this sample), and analyses have shown that the instrument needs further validation and tailoring to the target group. More research is needed to decrease comprehension problems and to investigate and retest reliability and construct validity [29].

## 5. Conclusions

Based on the present study’s findings, it can be concluded that children’s HL has an impact on some aspects of their lifestyle. A positive association between children’s HL and their PA behaviour was observed, while no significant association was found between children’s HL and their BMI z-score, vegetable, fruit, water, and soft drink consumption. The instrument used to measure children’s HL in the present study should be subjected to further refinement to increase its suitability for the target group (children). Additionally, comparable research in a larger, more diverse sample using more objective and sensitive data instruments to measure lifestyle behaviours and/or research investigating the association between children’s and parental HL is necessary to further advance the field of children’s HL and its relationship with their lifestyle.

## Figures and Tables

**Table 1 children-09-00925-t001:** Sample characteristics (*n* = 139).

Characteristic	N	%/mean (± SD)
Gender (% boys)	139	49.6
Age (years)	139	9.7 (0.656)
Grade	139	
*Grade three*	61	43.9
*Grade four*	78	56.1
Ethnicity	137	
*Dutch*	130	94.9
*Western*	4	2.9
*Non-Western*	3	2.2
SES (%) ^1^	139	
*Lowest tertile*	23	16.5
*Middle tertile*	45	32.4
*Highest tertile*	71	51.1
HL	139	3.1 (0.447)
BMI z-score	139	−0.2 (0.891)
PA score	139	3.0 (0.684)
Fruit consumption	139	
*Never*	1	0.7
*Almost never*	2	1.4
*Sometimes (1–3 times/week)*	13	9.4
*Often (4–6 times/week)*	43	30.9
*Every day*	80	57.6
Vegetable consumption	139	
*Never*	1	0.7
*Almost never*	2	1.4
*Sometimes (1–3 times/week)*	23	16.5
*Often (4–6 times/week)*	70	50.4
*Every day*	43	30.9
Water consumption	139	
*Never*	0	0.0
*Almost never*	11	7.9
*Sometimes (1–3 times/week)*	34	24.5
*Often (4–6 times/week)*	45	32.4
*Every day*	49	35.3
Soft drink consumption	139	
*Never*	6	4.3
*Almost never*	37	26.6
*Sometimes (1–3 times/week)*	62	44.6
*Often (4–6 times/week)*	17	12.2
*Every day*	17	12.2

Abbreviations: SD, standard deviation; SES, socio-economic status; HL, health literacy; BMI, body mass index; PA, physical activity. ^1^ Due to clustering of SES scores around several scores, the tertile group sizes are unequal.

**Table 2 children-09-00925-t002:** Relationship between children’s HL and BMI z-scores.

HL Measure	B (95% CI)	*p*
Mean HL scores	0.121 (−0.232; 0.473)	0.500
HL quintiles	0.124 (−0.127; 0.375)	0.331

Note. Analysed by multiple linear regression analyses. All analyses were adjusted for gender, age, and SES. Abbreviations: HL, health literacy; BMI, body mass index; CI, confidence interval.

**Table 3 children-09-00925-t003:** Relationship between children’s HL and PA summary scores.

HL Measure	B (95% CI)	*p*
Mean HL scores	0.356 (0.127; 0.585)	0.002 *
HL quintiles		
*Lowest HL*	−0.499 (−0.772; −0.225)	0.000 *
*Highest HL*	−0.019 (−0.267; 0.228)	0.879

Note. Analysed by linear mixed model analyses. All analyses were adjusted for gender, age, and SES. Abbreviations: HL, health literacy; PA, physical activity; CI, confidence interval. * Significant relationship.

**Table 4 children-09-00925-t004:** Relationship between children’s HL and fruit, vegetable, water, and soft drink consumption.

Outcome	HL Measure	B (95% CI)	*p*
Fruit consumption ^1^	Mean HL scores	0.151 (−0.119; 0.421)	0.271
	HL quintiles	−0.026 (−0.220; 0.167)	0.790
Vegetable consumption ^1^	Mean HL scores	0.167 (−0.104; 0.439)	0.225
	HL quintiles	0.132 (−0.061; 0.326)	0.179
Water consumption	Mean HL scores	0.242 (−0.132; 0.617)	0.203
	HL quintiles	0.155 (−0.113; 0.423)	0.254
Soft drink consumption	Mean HL scores	−0.202 (−0.601; 0.197)	0.319
	HL quintiles	−0.052 (−0.338; 0.234)	0.719

Note: Analysed by multiple linear regression analyses. All analyses were adjusted for gender, age, and SES. Abbreviations: HL, health literacy; CI, confidence interval. ^1^ For the outcomes of fruit consumption and vegetable consumption, three participants were excluded from the analyses due to outliers.

## Data Availability

The data presented in this study are available upon request from the corresponding author. The data are not publicly available as long as data collection in the overall research project is not completed.

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
