# Peer review of "Children’s Health Literacy in Relation to Their BMI z-Score, Food Intake, and Physical Activity: A Cross-Sectional Study among 8–11-Year-Old Children in The Netherlands"

_children, 2022, doi:10.3390/children9060925_

Round 1

Reviewer 1 Report

Thank you for submitting a good paper to Children. The study may be significant in that investigations of association between health literacy and lifestyle associated with body mass index in children. However, this study has some critical issues to be improved.

1.      Introduction is well organized, but too long. I recommend you to move description about other studies investigated relationship between HL and weight, food intake, and PA to the discussion.

2.      The sentence ‘there were no further inclusion or exclusion criteria.’ in the methods and the description of ‘3.1 sample’ seem contradiction. I recommend you to provide flowchart as figure 1.

3.      In general, physical activity is different between male and female. I recommend you to investigate the relationship between HL and physical activity after subdividing the participants into male and female.

4.      I recommend you to provide more evidence and theory that can support the relationship between HL and PA.

Reviewer 2 Report

General comments

The authors aimed to assess the association of Dutch children’s own Health Literacy with (1) their BMI z-score, (2) their dietary behaviour, and (3) the amount of Physical Activity they engage in. Children between the ages of 8 and 11 participated in the study. The authors hypothesized that children’s Helth Literacy would be negatively correlated with their weight and positively with their health-related behaviours.

The paper is well-organized and well-written. The topic is interesting.

I just have a few comments to make.

Specific comments

Abstract

It is clear and correctly summarizes the research work carried out.

Key Words

The keywords must be different from the title to allow you to have more chances when searching the web. Use different key words.

Introduction

It is well written. Good summary of the literature. The gap to be filled is clear. The aim and hypothesis are described correctly.

I would modify line 95 with "we aimed to" and line 98 with "It was hypothesized that children’s HL would be"

Materials and Methods

The methodology is explained correctly. Validated measurement tools and questionnaires were used. The statistical analyzes are correct.

To fix:

Line 120: Replace with 2.2.1

Line 138: Replace with 2.2.2

Line 149: Replace with 2.2.3

Results

Well-written section.

Line 223: Replace with “Table 3”

Discussion

The discussions are written clearly and to the point.

The limitations have been addressed.

The conclusions are justified.

The take-home message is clear.

Line 260: Replace with “would be negatively”
